# Diagnosis of Congenital Uterine Abnormalities: Practical Considerations

**DOI:** 10.3390/jcm11051251

**Published:** 2022-02-25

**Authors:** Kanna Jayaprakasan, Kamal Ojha

**Affiliations:** 1Derby Fertility Unit, Royal Derby Hospital, University of Nottingham, Derby DE22 3DT, UK; 2St. George’s Hospital, London SW17 0QT, UK; kojha2@gmail.com

**Keywords:** congenital uterine anomalies, diagnosis, 3D ultrasound, MRI, Mullerian duct, laparoscopy, hysteroscopy

## Abstract

As most congenital uterine abnormalities are asymptomatic, the majority of them are detected incidentally. While most women with uterine anomalies have a normal reproductive outcome, some may experience adverse reproductive outcomes. Accurate diagnosis and correct classification help in the appropriate counselling of women about their potential reproductive prognosis and risks and for planning any intervention. Evaluation of the internal and external contours of the uterus is the key in making a diagnosis and correctly classifying a uterine anomaly. Considering this, the gold standard test has been the combined laparoscopy and hysteroscopy historically, albeit invasive. However, 3D ultrasound has now become the diagnostic modality of choice for uterine anomalies due to its high degree of diagnostic accuracy, less invasive nature and it being comparatively less expensive. While 2D ultrasound and HSG are adequate for screening for uterine anomalies, MRI and combined laparoscopy and hysteroscopy are reserved for diagnosing complex Mullerian anomalies. Imaging for renal anomalies is recommended if a uterine anomaly is diagnosed.

## 1. Introduction

The process of reproduction including sperm transport, embryo implantation, foetal growth and development and the process of labour and birth rely on a structurally and functionally normal uterus. Any uterine abnormalities including congenital anomalies can influence some of these uterine functions adversely. While most women with uterine anomalies have a normal reproductive outcome, some may experience adverse reproductive outcomes, although the underlying pathophysiological process is uncertain.

Congenital uterine anomalies (CUA) are not uncommon, with the reported population prevalence rates in individual studies varying between 0.06% and 38%, and the observed wide variation is possibly due to the assessment of different study populations and the use of different diagnostic techniques [1]. Two well-conducted systematic reviews have evaluated the prevalence of uterine anomalies [1,2] with the more recent one by Chan et. al. (2011) reporting 5.5% in an unselected population, 8% in infertile women, 13.3% in those with a history of recurrent miscarriage and 24.5% in those with miscarriage and infertility.

CUAs result from embryological mal-development of the Mullerian ducts and depending on the degree and stages of Mullerian duct development, the types of CUAs vary. Therefore, knowing Mullerian duct embryology will help to understand the type and classification of CUAs. While this review primarily aims to discuss screening and correctly classifying CUAs using various diagnostic modalities, we will briefly discuss embryology and different classifications of uterine anomalies.

## 2. Development of Mullerian Ducts and Uterine Anomalies

The development of the female reproductive system involves a series of complex processes characterized by the differentiation, migration, fusion and canalization of the Mullerian duct system [3,4]. At around the 4th week of life, the primordial germ cells, developing among the endodermal cells in the dorsal wall of the yolk sac, migrate to the primitive gonads. Absence of the Y chromosome in female embryos differentiate the primordial germ cells to primary ovarian follicles. The lack of testosterone and AMH in a female foetus causes regression of the Wolffian ducts and allows the Mullerian ducts to develop into fallopian tubes, the uterus and upper third of the vagina (Figure 1). The fused Mullerian ducts join inferiorly with the urogenital sinus, causing proliferation of the endodermal sinovaginal bulbs, which form the vaginal plate. This then canalises to form the vagina with the upper third portion derived from the fused Mullerian duct and the other lower portion from the sinovaginal bulbs. Any deviation or failure of these steps mentioned above can lead to different types of CUAs (Table 1). Mullerian development is independent of gonadal development, and women with Mullerian anomalies usually have normal ovaries and ovarian function. Conversely, Mullerian development is closely associated with the development of the urinary tract, and therefore renal anomalies are often diagnosed in those with Mullerian anomalies. Simultaneous imaging of both the genital and urinary system are mandatory in women with suspicion of Mullerian anomalies.

While certain types of CUAs result from a defect in one specific stage of embryological development, some result from disturbances in more than one stage of normal development. The combination of abnormal development occurring at different stages of development is the reason for the observed wide anatomical variations and combined and complex congenital malformation of the female genital tract.

## 3. Classification of Uterine Anomalies

There are many classifications of CUAs published (Table 2) with the first of these reported by Cruveilher, Foerster and von Rokitansky in the mid-19th century. The descriptive classification introduced in 1979 by Buttram and Gibbons was later revised and modified by the American Fertility Society (AFS), now known as the American Society of Reproductive Medicine (ASRM) [5]. The AFS (ASRM) classification has been the most commonly used classification in the past and was based on the extent of failure of Müllerian development and divides anomalies into groups with similar clinical manifestations, management requirements and prognosis. The anomalies were classified as: hypoplasia/agenesis, unicornuate, didelphys, bicornuate, septate, arcuate and diethylstilboestrol (DES) drug-related. However, this classification is not without limitations; for example, it included only uterine anomalies with the exclusion of cervical and vaginal anomalies and did not classify combined or complex anomalies (obstructive-like cervical or vaginal abnormalities). There has been criticism on the lack of clear diagnostic criteria and also on the arcuate uterus being included as a separate class.

The European Society of Human Reproduction and Embryology (ESHRE) and the European Society for Gynaecological Endoscopy (ESGE), recognizing the clinical significance of Mullerian duct anomalies, have developed a new updated classification system through a structured Delphi procedure [6]. Uterine anomalies are classified, based on the anatomical deviations deriving from the same embryological origin, into seven main types: U0, normal uterus; U1, dysmorphic uterus (infantile or T-shaped); U2, septate uterus; U3, bicorporeal uterus (partial and complete—bicornuate and didelphys, respectively, based on AFS classification); U4, hemi uterus (unicornuate based on AFS); U5, aplastic uterus; U6, for unclassified cases (Figure 2). The arcuate uterus is considered as a normal variant and clinically irrelevant and is not included in this classification. Cervical and vaginal anomalies are classified as additional independent subclasses according to the increasing severity of the anatomical abnormalities (C0 to C4 and V0 to V4) with C0 and V0 being a normal cervix and vagina and C4 and V4 being cervical aplasia and vaginal aplasia, respectively. ESHRE/ESGE have attempted to define uterine anomalies based on 3D ultrasound measurements of uterine wall thickness and external and internal fundal indentations, although the defined uterine anomalies have not been correlated with reproductive outcomes. While it seems that the ESHRE/ESGE classification fulfilled the needs and expectations of a large group of experts in the field, it is not considered perfect due to a few reasons, including the reported increase in the diagnosis of septate uterus compared with former classifications as formerly classified normal or arcuate uteri are diagnosed as subseptate uteri under this classification system [8]. Based on a reproducibility and diagnostic accuracy study using 3D ultrasound, the Congenital Uterine Malformation Experts (CUME) group [9] has reported overestimating the prevalence of septate uterus if the ESHRE/ESGE classification is used.

Very recently, ASRM has revised the historic AFS (1988) classification following the formation of the ASRM task force on Mullerian anomalies classification (ASRM MAC 2021) (Table 3). They have expanded and updated the original AFS classification to include all the anomalies into nine distinct groups. They have recognised the Mullerian anomalies as a continuum of the variation in the embryological development and, therefore, some of the anomalies may be in more than one group, particularly for vaginal anomalies. However, the anomalies that are described by the ASRM MAC 2021 may still not include all known anomalies, given the unlimited number of possible variations.

## 4. Diagnosis

As most CUAs are asymptomatic, the majority of them are detected incidentally. A significant proportion of anomalies are diagnosed during fertility investigations. Accurate diagnosis and correct classification help in the appropriate counselling of women about their potential reproductive prognosis and risks and for planning any intervention with a view to improve the reproductive outcome. Evaluation of the internal and external fundal contours of the uterus is the key in making a diagnosis and correctly classifying a uterine anomaly. Considering this, the gold standard test has been the combined laparoscopy and hysteroscopy, albeit invasive, in the past. Imaging modalities such as ultrasonography, hysterosalpingogram (HSG), sonohysterogram and magnetic resonance imaging (MRI) are less invasive modes of screening and classifying various uterine anomalies [1]. While conventional 2D transvaginal ultrasound (TVS) and HSG are considered as good screening modalities, 3D TVS and MRI can accurately diagnose and classify the types of CUAs [10,11], as they can define both external and internal uterine contours.

### 4.1. Hysterosalpingogram

HSG helps in evaluating the uterine cavity but a definitive diagnosis of CUA requires evaluation of the external uterine contour, which is poorly defined by HSG. However, HSG, a commonly employed test to assess tubal patency as a part of fertility investigation, is a good screening test for CUAs. The angle of divergence of the uterine horns of less than 75° and more than 105° has been suggested to diagnose the septate and bicornuate uteri, respectively (Figure 3) [12]. However, the majority of these anomalies overlap and angles between the horns fall within this range. Furthermore, HSG cannot reliably differentiate between septate and bicornuate uteri because of its limitation in evaluating the external uterine contour. In addition, HSG can visualise the uterine cavity only if it communicates with the cervix and cases of non-communicating rudimentary uterine horn may be missed.

### 4.2. 2D Transvaginal Ultrasound

Conventional transvaginal ultrasound is minimally invasive and a less expensive way of assessing uterine morphology and ruling out uterine anomalies [13]. Ultrasound evaluation can be timed in the secretory (luteal) phase of the menstrual cycle as the endometrium, being bright and echogenic, is easy to visualise; ultrasound is therefore more appropriate for evaluating the uterus for congenital anomalies. The visualisation of a double endometrial complex on a transverse plane points towards a uterine anomaly (Figure 4) and the differential diagnosis would be a bicornuate, septate, subseptate or arcuate uterus. However, 3D ultrasound facilitates simultaneous visualization of both the external (serosal surface) and internal (endometrial) contours of the uterine fundus through its unique feature of providing the coronal plane of the uterus and can correctly classify the uterine anomaly into a bicornuate, septate or subseptate, or arcuate uterus [14].

Systematic scanning through the longitudinal plane of the uterus may reveal a uterine complex that then disappears while moving to the opposite side, followed by the appearance of a second uterine complex, suggesting that the uterus may be a partial or complete bicorporeal uterus (bicornuate or didelphys). The transverse plane provides more information and widely placed double endometrial echoes, especially at the upper portion of the uterus towards the fundus (Figure 4C), and an indentation at the fundus on an oblique plane (if obtainable) are typical of a bicornuate uterus. The double endometrial echoes will be closer in a septate uterus (Figure 4B) in contrast to that in a bicornuate uterus. The distance between the serosal surface and the upper border of the endometrial echo in a longitudinal plane may also give a clue to distinguish between septate and bicornuate uterus, although not confirmatory. In uterus didelphys, the two whole uterine body with endometrial echoes will be separate to each other and placed apart (Figure 4D), and the clinical demonstration of two cervices confirm the diagnosis. Two uterine horns may be symmetrical or asymmetrical and two separate vaginas may be seen on speculum examination.

A unicornuate uterus may be difficult to diagnose on a conventional 2D scan. On a longitudinal scan, a normal looking sagittal axis of the uterus is seen on one side in the pelvis with no or a rudimentary uterine shadow on the other side. A rudimentary or severely hypoplastic uterine horn is seen as an isoechoic pear-shaped structure with or without a central thin echogenic endometrial line (Figure 5). On the transverse plane, the uterus is tapered to one side and at the level of the fundus, a beak like projection from the endometrial shadow (uterine angle or shoulder from where the interstitial portion of the fallopian tube starts) is seen only on one side (Figure 5). A 3D ultrasound, again, is confirmatory, demonstrating a banana shaped uterine cavity with a single interstitial portion of the fallopian tube seen in the coronal plane (Figure 3). Saline infusion sonography may be helpful in diagnosing communicating rudimentary horns as saline can be clearly seen in the unicornuate uterus, with passage into the rudimentary horn.

### 4.3. Three-Dimensional (3D) Transvaginal Ultrasound

The 3D transvaginal ultrasound is considered the gold standard tool for the assessment of uterine anomalies as it is less invasive; it facilitates simultaneous visualization of both the external (serosal surface) and internal (endometrial) contours of the uterine fundus through its unique feature of providing the coronal plane of the uterus and, therefore, can correctly classify the types of uterine anomalies. While the method for diagnosis is agreed, there are different international ultrasound criteria reported. The criteria for the classification of uterine anomalies based on 3D ultrasound have been well described in the literature for the first time by Salim et al. in 2003 [14] (Table 4). ESHRE/ESGE have subsequently described the criteria in the Thessaloniki consensus on the diagnosis of female genital anomalies, although these have been criticized by some experts [8,9,15,16].

### 4.4. Technical Considerations

A suitable 3D US machine with a high frequency 3D transvaginal probe is used. The ultrasound setting is optimised to obtain a good quality image. The sweep angle is kept maximum (typically 120°) and the acquisition speed of maximum quality is selected.

The uterus is identified in the sagittal/longitudinal plane (unless it is a complete bicorporeal/didelphis uterus in which case it can be identified in the transverse plane to obtain the uterine complex in the same sweep). The depth of the window is such that the uterine body occupies at least three-quarters of the screen. The focus is placed at the level of the endometrial cavity. The reference plane is kept as the midsagittal plane of the uterus. The 3D function is activated and both the operator and the patient should remain completely still while the acquisition takes place. The acquisition should incorporate the entire uterine body(ies) [17].

Once the 3D acquisition is completed, the uterus is displayed on the screen in three orthogonal planes (A, B, C), which can be viewed in various different modes according to the US machine used (Figure 6 and Figure 7). The default display is in the sectional mode and the operator can choose the render mode, which enhances the contrast between the two areas by recreating an impression of depth and improves visual perception. The operators should familiarise themselves with manipulation of the uterus in the three planes using the X, Y and Z functions on the US console.

When the render mode is selected, a region of interest (ROI) box will appear on the screen. The ROI box should be manipulated so that it covers the endometrial cavity in its entirety. The green line of the ROI box indicates the direction of rendering and this should ideally be placed on top. Final adjustments can be made, such as curving the ROI box, along the curvature of the endometrial cavity and applying different render options, including Omniview, Volume Contrast Imaging (VCI) and HD live, which produces a more realistic image. Once the operator is satisfied with the 3D image, this can be enlarged and can correctly classify the type of CUA. The entire volume and individual image can be saved for later analysis. [18]. The post-processing of the stored image can be performed as described above on the ultrasound machine itself (commonly done) or using ‘4D view’ software in a computer (Figure 5).

### 4.5. Diagnostic Criteria Based on 3D Ultrasound

The diagnostic accuracy of 3D ultrasound compared with laparoscopy ± hysteroscopy in diagnosing CUA is highest among other imaging modalities including MRI. The diagnostic accuracy of 3D ultrasound is reported as 97.6% with sensitivity and specificity of 98.3% and 99.4%, respectively [18]. The morphology of the uterus is best examined in the coronal plane (Figure 8) using the interstitial portions of the fallopian tubes as reference points. A line joining the tubal ostia (interostial line) is the reference line (Figure 9). A parallel line on top of the fundus can be drawn and the vertical distance between this line and the interostial line is the uterine wall thickness. In cases of septate uterus, a parallel line along the apex of the internal midline indentation is drawn and its vertical distance from the interostial line is the septal length. The vertical distance between the parallel line along the bottom (apex) of the external indentation and interostial line is the depth of the external cleft in cases of partial bicorporeal (bicornuate) uterus.

The classification utilising 3D ultrasound based on the publication by Salim et al. (2003) is described in Table 2 [14]. The ESHRE/ESGE classification of uterine anomalies based on the 3D scan assessment is based on using uterine wall thickness as the reference. An internal indentation at the fundal midline of more than 50% of the uterine wall thickness is used to diagnose a septate uterus. On the other hand, a bicorporeal (bicornuate) uterus is diagnosed when the external indentation is more than 50% of the uterine wall thickness. The ASRM in the publication ‘Uterine septum: a guideline’ has suggested diagnosis of normal or arcuate uterus when the distance between the interostial line to the apex of indentation is less than 1 cm and the angle of indentation is more than 90° [19]. It also reported an arcuate uterus as a normal variant and clinically irrelevant. A septate uterus is diagnosed when the indentation depth is more than 1.5 cm and the angle of indentation less than 90°. A bicornuate uterus is diagnosed when the external fundal indentation is more than 1 cm.

The Congenital Uterine Malformation Experts (CUME) group has been critical of the ESHRE/ESGE criteria as overestimating and the ASRM criteria as underestimating the prevalence of septate uterus [9]. In a diagnostic accuracy study using 3D ultrasound, the proportion of septate uteri using the ESHRE/ESGE classification was demonstrated to be higher than using the ASRM criteria (RR 13.9; 95% CI 5.9–32.7, *p* ≤ 0.01) [20]. The concern about overdiagnosis is that it may lead to increased surgical intervention. The CUME group proposed a definition to diagnose septate uterus as an internal indentation of more than 10 mm, an angle of septal indentation of <140° and an indentation to the uterine wall thickness (I–WT) ratio of >110%. They proposed to use the septal indentation of >10 mm alone as the simplest and most reproducible criteria, if used alone. A summary of the criteria for diagnosing a septate uterus is detailed in Table 5.

While the T-shaped uterine cavity is diagnosed subjectively by some authors, the ESHRE/ESGE has suggested diagnosis based on a narrow uterine cavity due to thickened lateral walls without specifying a definition and cut-off for thickened lateral walls and narrow uterine cavity [21]. The CUME group proposed a lateral indentation angle ≤ 130°, lateral indentation depth ≥ 7 mm and T-angle ≤ 40° as the criteria for diagnosing a T-shaped uterus (Figure 10) [22].

### 4.6. Magnetic Resonance Imaging (MRI)

MRI has the multiplanar capability, allows better soft-tissue characterisation and permits a wider field of assessment at once than other imaging modalities including ultrasound [20]. MRI, due to its ability to demonstrate both the external and internal contours of the uterus, is sensitive and specific for diagnosing nearly all uterine anomalies. MRI is helpful in delineating the endometrium and detecting uterine horns regardless of the uterine position and anatomical variation. Furthermore, it is accurate in defining aberrant gonadal location or renal anatomy and is less invasive compared to laparo-hysteroscopy. In addition, it does have the ability to examine the details of the nearby structures including the cervix and vagina. While MRI is not routinely recommended in all women suspected to have a uterine anomaly, it proves useful for those patients with suspected complex anomalies, particularly if there is a combination of cervical and vaginal anomalies and for those at higher risk for associated anomalies. MRI is also useful if there is any diagnostic dilemma with other imaging modalities.

### 4.7. Renal Scan

Congenital uterine anomalies may be associated with congenital renal anomalies because of closely related embryogenesis. In a recent study of 378 women with uterine anomalies [23], renal abnormalities were found in 18.8%, unilateral renal agenesis being the most common defect (12.2%). When different subtypes based on the ESHRE/ESGE criteria were assessed, the prevalence of renal anomalies in normal (U0), dysmorphic (U1), septate (U2), bicorporeal (U3), hemi uterus (U4) and aplastic (U5) were 5%, 0%, 15.6%, 24.7%, 29.5% and 11.7%, respectively. An abdominal ultrasound scan or MRI is recommended in all women diagnosed to have uterine anomalies.

## 5. Conclusions

Uterine anomalies are commonly seen in women presenting with a history of reproductive problems. While 2D ultrasound and HSG are adequate for screening for uterine anomalies, 3D ultrasound, MRI and combined laparoscopy and hysteroscopy can correctly classify the type of uterine anomaly due to their ability to show both external and internal contours of the uterus. While 3D ultrasound is now considered as the gold standard diagnostic tool for uterine anomalies due to its high degree of diagnostic accuracy, less invasive nature and it being comparatively less expensive, MRI is reserved for diagnosing complex Mullerian anomalies or if there is a diagnostic dilemma. Laparoscopy and hysteroscopy are an invasive modality for diagnostic purposes and should be offered only in the context of concomitant surgical treatment after a thorough non-invasive evaluation of a Mullerian anomaly. Imaging for renal anomalies is recommended if a uterine anomaly is diagnosed.

## Figures and Tables

**Figure 1 jcm-11-01251-f001:**
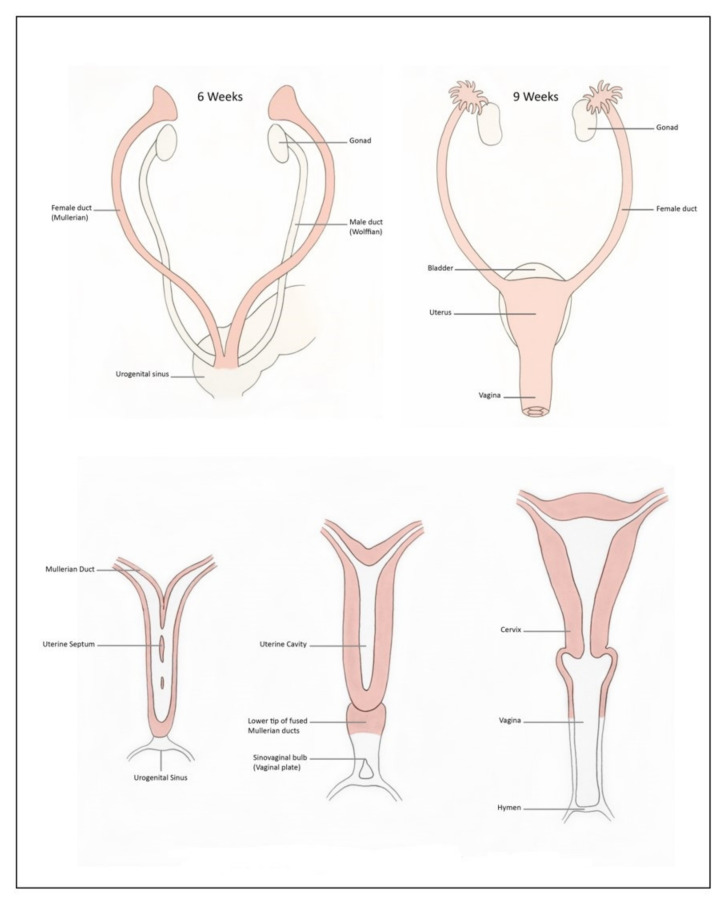
Development of uterus and vagina (adapted from [3]).

**Figure 2 jcm-11-01251-f002:**
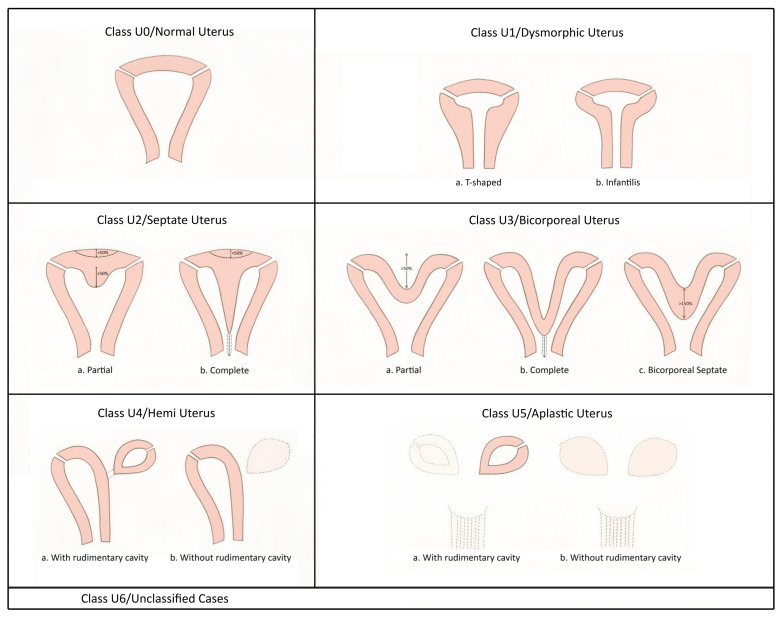
ESHRE/ESGE classification of uterine anomalies (adapted from [6]).

**Figure 3 jcm-11-01251-f003:**
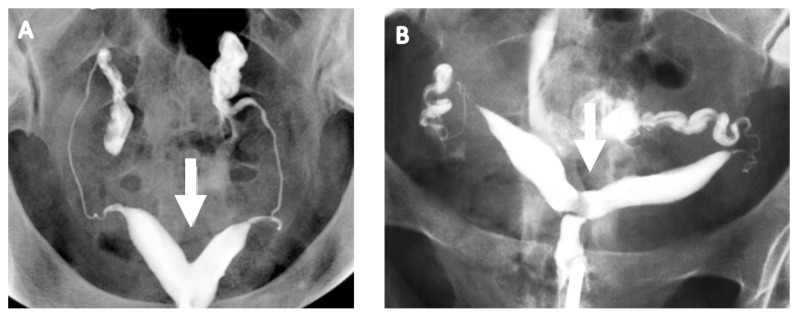
Hysterosalpingogram showing acute angle in a septate uterus (**A**) and wider angle in a bicornuate uterus (**B**).

**Figure 4 jcm-11-01251-f004:**
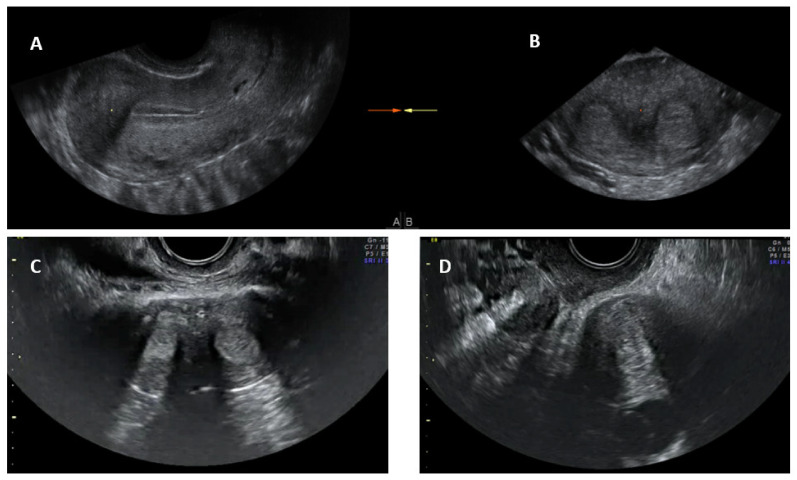
Longitudinal section of subseptate uterus in midsagittal plane (**A**); transverse plane of a subseptate uterus showing two endometrial echoes (**B**); transverse plane of a bicornuate uterus showing two endometrial echoes (**C**); transverse plane of uterus didelphys showing two uterine bodies (**D**).

**Figure 5 jcm-11-01251-f005:**
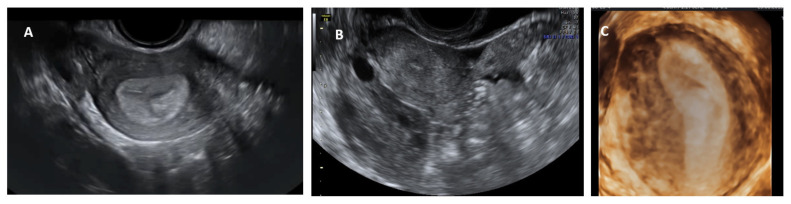
Unicornuate uterus: 2D transverse view showing only one uterine angle (shoulder)—Right (**A**); a small left rudimentary uterine horn (**B**) and banana-shaped uterine cavity on a 3D coronal plane (**C**).

**Figure 6 jcm-11-01251-f006:**
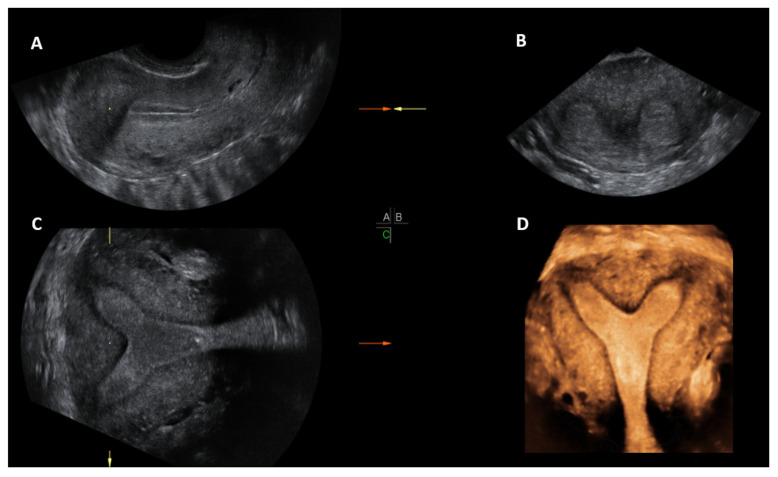
3D ultrasound scan of a subseptate uterus showing simultaneous display of longitudinal plane (**A**), transverse plane showing two endometrial echoes (**B**), coronal plane (**C**), unique for 3D ultrasound and Rendered view of coronal plane demonstrating subseptate uterus (**D**).

**Figure 7 jcm-11-01251-f007:**
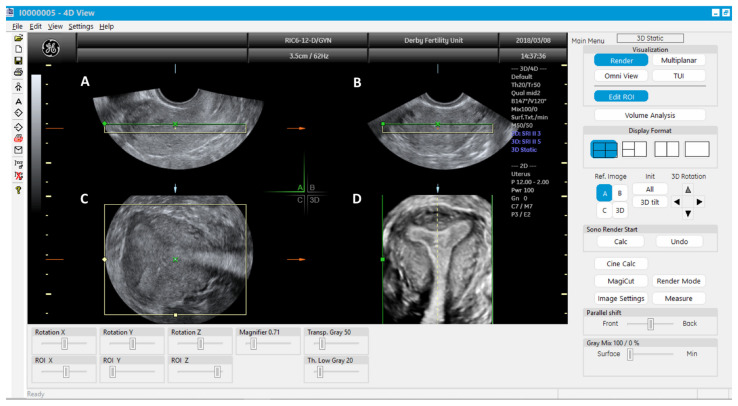
3D multiplanar view with rendering box using 4D view software: Green line of the rendering box placed on top and at the level of endometrial cavity in the longitudinal plane (**A**); transverse plane (**B**), Coronal plane (**C**) and rendered coronal view of uterus in the bottom right corner (**D**).

**Figure 8 jcm-11-01251-f008:**
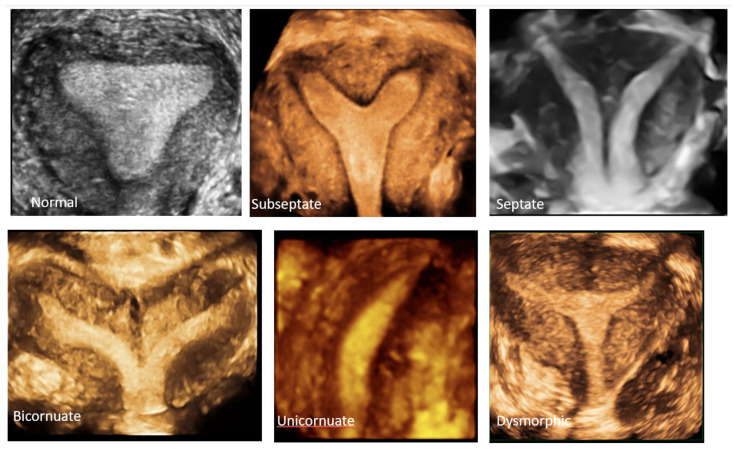
3D coronal plane of uterus: normal uterus, subseptate uterus, septate uterus, bicornuate, unicornuate uterus and dysmorphic (T-shaped) uterus.

**Figure 9 jcm-11-01251-f009:**
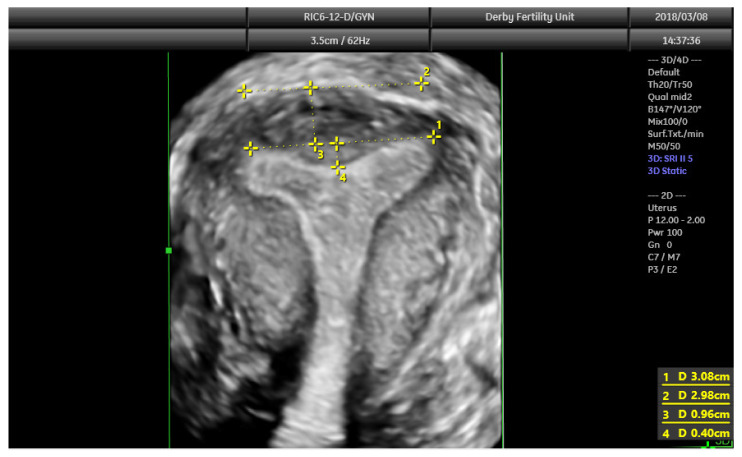
3D coronal plane of uterus with assessments: interostial line (measurement 1); a parallel line along the serosal surface (measurement 2); uterine wall thickness (measurement 3) and septal indentation length. This uterus is not septate, but may be classified as arcuate uterus, which has no clinical relevance based on the recent ESHRE and ASRM guidelines.

**Figure 10 jcm-11-01251-f010:**
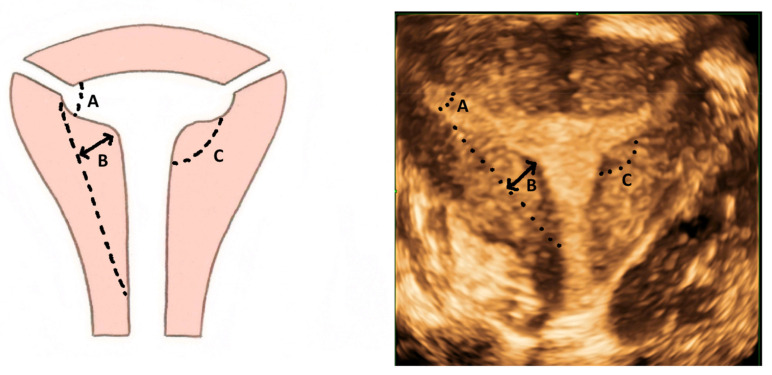
Criteria for diagnosing T-shaped uterus according to CUME ([22]). A—T-angle ≤ 40°; B—lateral indentation depth ≥ 7 mm; C—lateral indentation angle ≤ 130°.

**Table 1 jcm-11-01251-t001:** Phases of Mullerian duct development and defects.

Phases of Mullerian Duct Development	Defect	Anomaly
**Organogenesis:** Development of Mullerian Duct	Failure to develop bilaterally	Aplasia/agenesis (MRKH * syndrome)
	Failure to develop unilaterally	Unicornuate uterus
**Fusion or Unification:** between paired Mullerian ducts between fused Mullerian duct and urogenital sinus (Sinovaginal bulbs)	Horizontal fusion defect	Bicornuate uterus Uterus didelphys
	Vertical fusion defect	Transverse vaginal septum Imperforate hymen
**Septal resorption or canalisation**	Defect in resorption or canalisation	Septate uterus Arcuate (?)

* Mayer–Rokitansky–Küster–Hauser syndrome.

**Table 2 jcm-11-01251-t002:** Main classification systems of Mullerian anomalies and characteristics.

Guideline	Main Characteristics
AFS (1988) [5]	Based on failure of normal Mullerian development (hypoplasia/agenesis, unicornuate, bicornuate, didelphys, septate, arcuate, DES drug-related), lacked clear diagnostic criteria, correlated with clinical outcome
ESHRE-ESGE (2013) [6]	Classified based primarily on uterine anatomy with cervical vaginal anomalies as supplementary subclasses. Classes: U0–U6 (U0 being normal or arcuate and U6 being unclassified) C0–C4, V0–V4 3D-based diagnostic criteria for septate and bicornuate uterus
ASRM (2021) [7]	Updated and expanded AFS (1988) classification incorporating cervical, vaginal and all complex anomalies. Nine classes: Mullerian agenesis, cervical agenesis, unicornuate, uterus didelphys, bicornuate, septate, longitudinal vaginal septum, transverse vaginal septum, complex anomalies. Imaging-based diagnostic criteria for septate and bicorporeal uterus.

**Table 3 jcm-11-01251-t003:** ASRM Mullerian anomalies’ classification 2021 [7].

Main Category	Subcategories
Mullerian agenesis	- Complete Müllerian agenesis - Müllerian agenesis with R/L atrophic uterine remnant with functional endometrium
Cervical agenesis	- Complete Cervical agenesis - Distal Cervical agenesis
Unicornuate uterus	- R/L Unicornuate uterus - R/L Unicornuate with R/L distal atrophic uterine remnant - R/L Unicornuate with R/L distal uterine remnant with functional endometrium - R/L Unicornuate with R/L associated atrophic uterine remnant - R/L Unicornuate with R/L uterine horn communicating at level of cervix
Uterus Didelphys	- Uterus didelphys and complete longitudinal vaginal septum - Uterus didelphys and +/− longitudinal vaginal septum of variable length - Uterus didelphys and obstructed R/L hemi vagina
Bicornuate uterus	- Bicornuate uterus (with single cervix) - Bicornuate uterus with R/L communicating tract - Uterus bicornuate bicollis - Combined bicornuate septate uterus
Septate uterus	- Partial septate uterus - Normal/arcuate uterus - Robert’s uterus (Septate uterus with non-communicating hemi uterus) - Complete septate uterus with duplicated cervices and longitudinal vaginal septum - Complete septate uterus with septate cervix and longitudinal vaginal septum - Complete septate uterus, duplicated cervices, and obstructed R/L hemi vagina
Transverse vaginal septum	- Midvaginal septum - Distal vaginal agenesis
Longitudinal vaginal septum	- Longitudinal vaginal septum of variable length - Longitudinal vaginal septum of variable length and uterus didelphys - Longitudinal vaginal septum of variable length and complete septate uterus with duplicated cervix - Obstructed R/L hemi vagina and uterus didelphys - Obstructed R/L hemi vagina and complete septate uterus with duplicated cervices
Complex anomalies	- Bicornuate uterus with bilateral obstructed endometrial cavities - Uterus didelphys with communicating hemi uteri and unilateral R/L cervico-vaginal atresia - Obstructed R/L hemi vagina, hemi uterus and single cervix with separate contralateral R/L patent hemi uterus, cervix and vagina - Bicornuate uterus with R/L communicating tract and transverse vaginal septum - Uterus isthmus agenesis

**Table 4 jcm-11-01251-t004:** Classification of uterine anomalies based on 3D ultrasound assessment (Salim et al. 2003) [14].

Uterine Morphology	Internal Contour	External Contour
Normal	Straight or convex	Uniformly convex or with indentation < 10 mm
Arcuate	Concave fundal indentation with central point of indentation at obtuse angle (>90°)	Uniformly convex or with indentation < 10 mm
Subseptate	Presence of septum, which does not extend to cervix, with central point of septum at an acute angle (<90°)	Uniformly convex or with indentation < 10 mm
Septate	Presence of uterine septum that completely divides cavity from fundus to cervix	Uniformly convex or with indentation < 10 mm
Unicornuate	Single well-formed uterine cavity with a single interstitial portion of fallopian tube and concave fundal contour	Fundal indentation > 10 mm dividing the two cornua if rudimentary horn present
Bicornuate	Two well-formed uterine cornua	Fundal indentation > 10 mm dividing the two cornua
T-Shaped	T-shaped uterine cavity	

**Table 5 jcm-11-01251-t005:** Diagnostic criteria for septate uterus as described by various classification systems.

Guideline	Diagnostic Criteria
AFS (1988) [5]	Subjective impression and clinically relevant
ESHRE/ESGE (2013) [6]	Indentation (I) to wall thickness (I–WT) ratio > 50%
ASRM (2016) [19]	Septal angle < 90° and Septal length > 15 mm
CUME (2018) [9]	Septal length > 10 mm, Septal angle < 140° and I–WT > 110%
ASRM (2021) [7]	Septal angle < 90° and Septal length > 10 mm

## Data Availability

Not applicable.

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
