# Peer review of "Diagnosis of Congenital Uterine Abnormalities: Practical Considerations"

_jcm, 2022, doi:10.3390/jcm11051251_

Round 1
Reviewer 1 Report
Important points:
1. Include T-shaped uterus
2. Create a table with the suggested diagnostic criteria (ASRM 2021, ESHRE, and CUME), preferentialy commenting on the differences.
Some important references:
- Pfeifer SM, Attaran M, Goldstein J, Lindheim SR, Petrozza JC, Rackow BW, Siegelman E, Troiano R, Winter T, Zuckerman A, Ramaiah SD. ASRM müllerian anomalies classification 2021. Fertil Steril. 2021 Nov;116(5):1238-1252. doi: 10.1016/j.fertnstert.2021.09.025. PMID: 34756327.
- Coelho Neto MA, Ludwin A, Petraglia F, Martins WP. Definition, prevalence, clinical relevance and treatment of T-shaped uterus: systematic review. Ultrasound Obstet Gynecol. 2021 Mar;57(3):366-377. doi: 10.1002/uog.23108. PMID: 32898287.
- Ludwin A, Coelho Neto MA, Ludwin I, Nastri CO, Costa W, Acién M, Alcazar JL, Benacerraf B, Condous G, DeCherney A, De Wilde RL, Diamond MP, Emanuel MH, Guerriero S, Hurd W, Levine D, Lindheim S, Pellicer A, Petraglia F, Saridogan E, Martins WP. Congenital Uterine Malformation by Experts (CUME): diagnostic criteria for T-shaped uterus. Ultrasound Obstet Gynecol. 2020 Jun;55(6):815-829. doi: 10.1002/uog.20845. Epub 2020 May 15. PMID: 31432589.
- Ludwin A, Martins WP, Bhagavath B, Lindheim SR. Overdiagnosis, overdetection, and overdefinition of the septate uterus: reexamination of the ASRM and ESHRE-ESGE criteria is urgently needed. Fertil Steril. 2019 Sep;112(3):448-449. doi: 10.1016/j.fertnstert.2019.05.026. Epub 2019 Jul 29. PMID: 31371057.
Author Response
Thank you very much for your valuable comments, suggestions and also providing relevant references.
We have already included T- shaped uterus, now have added diagnostic criteria as suggested. We have now added ASRM 21 classification along with a table for ASRM classification. To include all the classification in a single table can be complex, however.
Reviewer 2 Report
1) The Figures 1 and 2 are reproduced with no comment of obtained permission.
2) I cannot find any novelty in the paper submitted. On the contrary I've found many mistakes.
The information in the chapter “Development of Mullerian ducts and uterine anomalies” are very basic.
The authors mix various classification systems and are inconsequent.
Saline infusion sonography is not a method useful for diagnosing rudimentary uterine horn.
HSG performed in women with infertility to verify tubal patency may suggest uterine anomaly but it is not a screening procedure.
Laparoscopy and hysteroscopy are invasive techniques and should not be used for diagnosing. These techniques should be offered only in the context of concomitant surgical treatment after a thorough non-invasive evaluation of discovered pathology.
Author Response
Thanks very much for your comments and suggestions. We have gone through and ensured that the contents are correct with some edits. We believe that the edits that we made have improved the manuscript to be useful for the readers. With respect to the the development of the Mullerian ducts, we want it to be basic considering the context and intended content of the article
Figure 1 and 2 are re-drawn by ourselves and not copied directly from the published journals. Therefore, we thought we wouldn't need permission. However, if need permission from the publishers for re-drawing, we can obtain that.
We have reported all the classifications that are published for the readers' knowledge. Now that we have added two further classifications as per reviewer1's suggestions.
We think HSG is a 1st line screening tool albeit not as a routine. But for patients who had HSG for fertility reasons, HSG is helpful in diagnosing uterine anomalies. We have now clarified that.
With respect to saline infusion sonography - we have now modified to make it clearer -"Saline infusion sonography has been suggested as a method for diagnosing non-communicating rudimentary horns as saline can be clearly seen in the unicornuate uterus, with no passage into the rudimentary horn."
With respect to laparoscopy hysteroscopy - We have modified as suggested - "Laparoscopy and Hysteroscopy are invasive modality for diagnostic purpose and should be offered only in the context of concomitant surgical treatment after a thorough non-invasive evaluation of Mullerian anomaly."
Reviewer 3 Report
Line 50. Absence of the sex determining region (SRY) gene in female embryos allows the gonads to develop into ovaries.
You have to change it to: absence of the Y chromosome in female embryos differentiate the primordial germ cells to primary ovarian follicles
Line 53: Change to : Upper third of the vagina.
Line 55: the upper portion derived from the fused Mullerian duct and lower portion from the sinovaginal bulbs. You have to change it to: the upper third portion derived from the fused Mullerian duct and the rest lower portion from the sinovaginal bulbs
Line 62: It would be appropriate to add a comment such as: which makes the simultaneous imaging investigation of these two systems mandatory
Line 107: based on latest bibliography we know that a large percentage of these abnormalities are discovered during infertility investigation and you may add a short comment Line 177 : 4.3.3. D. Change to 4.3 3D Comments should be added in relation to the training of infertility specialists and general gynecologists in the practice of conventional 2D and specialized 3D ultrasound who may accidentally find themselves in front of such a controversial finding, so that it is possible to reproduce and use the technique correctly.

Author Response
Thanks very much for your comments and valuable suggestions. We have incorporated all those and amended the article.
Round 2
Reviewer 1 Report
Having a table including only the diagnostic criteria from one study (Salim 2003) is not appropriate, as this will make readers select these cut-off points into their clinical practice. Authors should either present all the current diagnostic criteria or none.
Author Response
Thank you for re-reviewing the article. As the reviewer suggested, we have now added three more tables including one for diagnostic criteria for uterine septum. We have incorporated Salim et. al. 2003 paper, which is the first study describing criteria for diagnosis based on 3D scan
Reviewer 2 Report
The manuscript has been improved.
Page 9: Saline infusion sonography may be helpful in diagnosing COMMUNICATING rudimentary horns as saline can be clearly seen in the unicornuate uterus WITH PASSAGE into the rudimentary horn. If no passage is observed discrimination between non-communicating rudimentary horn and no rudimentary horn cannot be made with saline infusion sonography.
First paragraph Page 5 and second paragraph page 13 are identical. I suggest deletion of one of them.
Fig 1 and 2: Appropriate citations or the proof the reproduction of the figure is permitted must be added.
Author Response
Thank you for re-reviewing the revised article.
Thank you for suggesting the changes and we amended as "Saline infusion sonography may be helpful in diagnosing COMMUNICATING rudimentary horns as saline can be clearly seen in the unicornuate uterus WITH PASSAGE into the rudimentary horn. If no passage is observed discrimination between non-communicating rudimentary horn and no rudimentary horn cannot be made with saline infusion sonography."
First paragraph Page 5 and second paragraph page 13 are identical. I suggest deletion of one of them.
Thank you for pointing out duplication paragraph on page 5 and 13 and now the sentence on page 5 have been deleted.
Fig 1 and 2: Appropriate citations or the proof the reproduction of the figure is permitted must be added.
As re-drawn, we have appropriately referenced
This manuscript is a resubmission of an earlier submission. The following is a list of the peer review reports and author responses from that submission.